# Ozone in Chemotherapy-Induced Peripheral Neuropathy—Current State of Art, Possibilities, and Perspectives

**DOI:** 10.3390/ijms24065279

**Published:** 2023-03-09

**Authors:** Katarzyna Szklener, Anna Rudzińska, Pola Juchaniuk, Zuzanna Kabała, Sławomir Mańdziuk

**Affiliations:** Department of Clinical Oncology and Chemotherapy, Medical University of Lublin, 20-954 Lublin, Poland

**Keywords:** CIPN, ozone, pain treatment

## Abstract

Chemotherapy-induced peripheral neuropathy (CIPN) is one of the most detrimental toxicity to a patient’s quality of life. Pathophysiological mechanisms involved in CIPN pathogenesis are complex, multifactorial, and only partially examined. They are suspected to be associated with oxidative stress (OS), mitochondrial dysfunction, ROS-induced apoptosis, myelin sheath and DNA damage, and immunological and inflammatory processes. Unfortunately, medications commonly used for the management of other neuropathic pain syndromes, including gabapentinoids, opioids, and tricyclic antidepressants (such as desipramine and nortriptyline), do not bring satisfactory results in CIPN. The aim of this review is to evaluate the existing literature on the potential use of medical ozone as a treatment for CIPN. This paper would explore the potential therapeutic benefits of medical ozone. The review would evaluate the existing literature on the use of medical ozone in other contexts, as well as its potential application in treating CIPN. The review would also suggest possible research methods, such as randomized controlled trials, to evaluate the efficacy of medical ozone as a treatment for CIPN. Medical ozone has been used to disinfect and treat diseases for over 150 years. The effectiveness of ozone in treating infections, wounds, and a variety of diseases has been well documented. Ozone therapy is also documented to inhibit the growth of human cancer cells and has antioxidative and anti-inflammatory effects. Due to its ability to modulate oxidative stress, inflammation, and ischemia/hypoxia, ozone may have a potentially valuable effect on CIPN.

## 1. Introduction

Chemotherapy (CT) remains one of the mainstays of cancer treatment. Due to the advances in medicine and technology, its efficacy is growing and overall cancer mortality has decreased [1].

Unfortunately, oncological patients and cancer survivors have to face multiple unpleasant adverse consequences of using antineoplastic agents. This is caused by the fact that chemotherapeutics work by targeting rapidly dividing cancerous cells, but they also affect proper functioning, healthy cells. One of the most common mechanisms that underlie the causes of chemotherapy-induced toxicity involves loss of homeostatic control of reactive oxygen species (ROS) [2].

For some drugs, the most detrimental toxicity to a patient’s quality of life is chemotherapy-induced peripheral neuropathy (CIPN), which can occur in 19% to more than 85% of patients. Incidence of this condition varies among the different classes of agents. The risk of such impairment of the peripheral nervous system is the highest during platinum-based drug therapy—between 70% and 100% [3].

Pathophysiological mechanisms involved in CIPN pathogenesis are complex, multifactorial, and only partially examined. They are suspected to be associated with oxidative stress (OS), mitochondrial dysfunction, ROS-induced apoptosis, myelin sheath and DNA damage, and immunological and inflammatory processes [3,4,5].

CIPN displays sensory symptoms more often than motor symptoms. Its manifestation often includes paresthesia, dysesthesia, hypoalgesia or pain (burning, shooting or electric-shock-like), and cramping of the extremities [6,7]. Symptoms are symmetric and typically spread from distal to proximal level. With lower prevalence, the perioral region can be affected which leads to laryngospasm [6].

Painful symptoms sometimes persist well beyond the discontinuation of treatment [6,7].

The clinical presentation of this condition typically begins with symptoms in the feet and hands, which then progress proximally to involve the ankles and wrists in a characteristic “stocking and glove” distribution. One hallmark symptom of the condition is a loss of sensitivity to heat, which occurs due to selective damage to myelinated primary afferent sensory nerve fibers, with or without accompanying demyelination [7].

Local alterations may aggravate general cancer-related symptoms, such as anxiety, depression, or fatigue.

The development of CIPN may be a serious obstacle in chemotherapy because of its considerable impact on the patient’s quality of life. CIPN can lead to dose reduction and even cessation of CT which may have a direct contribution to its effectiveness [4].

Unfortunately, medications commonly used for the management of other neuropathic pain syndromes do not bring satisfactory results in CIPN [8]. Further research for the suitable treatment for CIPN is needed because of its unclear pathogenesis. However, various pharmacological and nonpharmacological treatment approaches (i.v. lidocaine, cannabinoids, oral glutamine, cryotherapy, acupuncture, massage therapy, and exercise) are being considered as possible CIPN management options. Some of them show tremendous potential but require further clinical studies [9]. The only agent mentioned in the recommendations that proved to be effective against CIPN is duloxetine [10]. While the exact mechanisms by which duloxetine exerts its therapeutic effects in CIPN are not fully understood, it is thought to act in part by reducing inflammation and oxidative stress in the nerves. Therefore, the therapies that take into account the OS role in CIPN pathogenesis may appear to be promising [11].

Ozone (O3) is a gas discovered in the mid-19th century and was initially used as a potent disinfectant. Its molecule contains three atoms of oxygen in a dynamically unstable structure due to the presence of mesomeric states [11,12]. The gas is colorless, acrid in odor, and explosive in liquid or solid form. It has a half-life of 40 min at 20 °C and approximately 140 min at 0 °C. Its primary function is to shield humans from the harmful effects of UV radiation. Ozone has been demonstrated to be toxic to both animals and humans, irritating the eyes and respiratory tract. Inhaling higher concentrations of ozone may cause damage to the bronchial mucosa and pneumocytes, leading to pulmonary edema. It has been calculated that breathing pure ozone at a concentration of 0.02 μg/mL leads to death in 4 h. There have been no other toxic effects discovered. It should be noted that the main gasses in the air we breathe, oxygen, nitrogen, and carbon dioxide, are also toxic and lethal if inhaled in high concentrations [12]. Ozone concentration is measured in μg/mL or g/L of oxygen; 5%, or 70 μg/mL, is usually the maximum concentration used in clinical medical applications. High concentrations will damage red blood cells and inhibit the growth of healthy cells [13]. The best technology for producing ozone gas was designed and built by Nikola Tesla in the 1920s. Medical ozone is a 5% ozone and 95% oxygen mixture obtained from medical oxygen by using a medical device—a medical ozone generator [14,15]. Generators use several techniques to produce ozone, such as UV lamps, corona discharge, and cold plasma generators [13].

Medical ozone has been used to disinfect and treat diseases for over 150 years. The effectiveness of ozone in treating infections, wounds, and a variety of diseases has been well documented. A text on medical ozone therapy was published by Dr. Charles J. Kenworth in 1885 [13]. Ozone has been shown to have numerous therapeutic benefits due to its antibacterial, antiviral, and antifungal properties [16]. Ozone therapy is also documented to inhibit the growth of human cancer cells and has antioxidative and anti-inflammatory effects [17]. The administration of ozone therapy varies depending on the treatment goals and location of the therapy: direct i.v. infusion, major autohemotherapy, rectal/vaginal insufflation, limb or body bagging, ozonated water, ozone in saline or LRS, intramuscular, intradiscal, paravertebral, intra-articular administration, prolo/sclerotherapy, acupuncture, ozonated olive oil, inhalation (ozone bubbled through olive oil and humidified), subconjunctival injection, gingival and tooth apex injection, urinary bladder insufflation, and auricular [13,18]. Ozone’s clinical use may be classified into eight categories: cardiovascular (coronary artery disease and previous myocardial infarction), subcutaneous tissue (diabetic foot ulcer, Buruli ulcer, nonhealing or ischemic wound), peripheral vascular disease (obliterative atheromatosis and peripheral artery disease), neurological (multiple sclerosis, refractory headache, and radiation-induced brain ischemia), head and neck (sensorineural hearing loss, head and neck tumors, vestibulocochlear syndrome, and dry form of age-related macular degeneration), orthopedic (herniated lumbar discs, spine and joint osteoarthritis, first-degree spondylolisthesis, and spondylolysis) gastrointestinal (chronic hepatitis C, liver cirrhosis, and gastrointestinal tract ulcers), and genitourinary (chronic cystitis, renal complications secondary to hepatitis, radiation-induced cystitis with hematuria, and urinary tract infection). The indications listed above are the results of human clinical trials for specific pathologies associated with the aforementioned systems [18]. Despite existing evidence that ozone therapy is safe and effective, more research is needed to establish it as an essential treatment option in medicine.

## 2. Methodology

In the light of possible ozone CIPN treatment, the revelations describing the usage of ozone in the treatment of pain pathology stemming from nerve damage and pain conductivity dysfunction hold significant value.

The aim of this review is to evaluate the existing literature on the potential use of medical ozone as a treatment for chemotherapy-induced peripheral neuropathy (CIPN), a condition that can severely impact a patient’s quality of life. The majority of the existing literature focuses on the complex and multifactorial pathophysiological mechanisms that contribute to the development of CIPN, including oxidative stress, mitochondrial dysfunction, inflammation, and DNA damage.

This review is to explore the potential therapeutic benefits of the potential application of medical ozone in treating CIPN. The methodology involves a comprehensive search of relevant databases (PubMed, Google Scholar), using appropriate keywords related to the theme (CIPN/oxidative stress/pain reduction/pain management and ozone therapy). All identified publications on the use of ozone therapy for CIPN treatment or pain management were included. Data from selected articles have been presented as a classical review. Identified clinical trial results have been provided in Table 1, while the reported studies on neuropathic and cancer-related pain treated with ozone have been summarized in Table 2.

The main limitation of the review is the relatively small amount of available literature, including randomized clinical trials. There is much more information about the treatment of pain with ozone than about its use in induced neuropathic pain treatment. In addition, the poorly understood pathophysiology of CIPN results in a lack of a solid theoretical basis for the assessment of the collected material, which increases the risk of bias.

## 3. Pathophysiology and Molecular Characteristics of CIPN

Classical antitumoral drugs are widely known for their cytotoxicity and prominent side effects, especially in fast-dividing cells such as those of the bone marrow, GI tract, reproductive system, and hair follicles [30]. Even though nervous tissue is not rapidly proliferating, antitumoral drugs may cause neurotoxicity, both directly and indirectly, causing sensory symptoms that include hyperalgesia, allodynia, spontaneous shooting or burning pain, dysesthesia, paresthesia, and other deficits in not only sensory but also autonomic and motor function [31]. Neurotoxicity induced by chemotherapy mainly affects the peripheral nervous system (PNS) due to its lack of protection from a structure similar to the blood–brain barrier (BBB) that protects the central nervous system (CNS) [32]. The longer the axon, the more vulnerable it is to the toxicity of chemotherapeutic agents which may be caused by their higher metabolic requirements [33].

The effects of anticancer drugs on the nervous system depend on their physical and chemical properties and their dosage, so they vary across the different classes of chemotherapeutics [32]. Research on CIPN from the past 20 years points to four main mechanisms of antitumor drugs: (1) directly targeting the mitochondria and producing oxidative stress; (2) functionally impairing ion channels; (3) triggering immunological mechanisms through activation of satellite glial cells; and/or (4) disruption of microtubules [34]. There are six main substance groups that contribute to the development of CIPN: the platinum-based antineoplastics (oxaliplatin and cisplatin), the vinca alkaloids (vincristine and vinblastine), the epothilones (ixabepilone), the taxanes (paclitaxel, docetaxel), the proteasome inhibitors (bortezomib), and the immunomodulatory drugs (thalidomide) [7].

### 3.1. Oxidative Stress in CIPN Development

Many antineoplastic agents are known for their ability to cause oxidative stress, which is the imbalance between the production of ROS and the ability to detoxify their detrimental effects [35]. Both bortezomib and paclitaxel can increase the production of ROS, which is also produced in small amounts in healthy tissues as a by-product of oxygen metabolism but in excess may worsen mitochondrial function [36]. Overproduction of ROS can lead to damage of intracellular biomolecules such as phospholipids, which results in demyelination, oxidation of proteins, releasing carbonyl by-products that can sensitize TRPV channels, inactivate antioxidant enzymes, and damage microtubules [37]. ROS can also indicate the activation of apoptotic pathways and the overproduction of pro-inflammatory mediators [38].

The role of oxidative stress and mitochondria dysfunction in the pathobiology of CIPN is supported by many in vitro and in vivo studies [33]. Observations of sectioned peripheral nerves in rodents previously treated with anticancer drugs show swollen and vacuolated mitochondria [4,10]. Preclinical studies on strategies targeting ROS based on external antioxidant stimulation were promising but it did not translate into clinical studies. With the results of many studies on antioxidants such as α-lipoic acid being disputed, endogenous antioxidants and peroxisome-proliferator-activated receptors (PPARs) were proposed as potential targets for in vivo CIPN studies [33,36]. Substantial evidence shows that further study of 4-hydroxy-2-nonenal (4-HNE), a secondary intermediate of oxidative stress and one of the most formidable reactive aldehydes, and the mechanisms of its regulation may be applicable to many oxidative-stress-related injuries. Furthermore, it is to be expected that other molecular targets, such as aldehyde dehydrogenase (ALDH2) and Alda-1, a selective activator of ALDH2, will be increasingly studied [39].

The transient receptor potential ankyrin 1 (TRPA1) channel is a major oxidant sensor profusely expressed by a subpopulation of primary sensory neurons. It is proposed that, during the therapy with some antitumoral drugs such as thalidomide, platinum-based antitumor drugs, vinca alkaloids, and paclitaxel pain may be induced through the upregulation of its expression [5].

### 3.2. Platinum-Induced Neurotoxicity

Platinum-based drugs are an important part of chemotherapy which are used to treat different types of solid tumors; however, PT chemotherapy is not tumor-specific and always affects normal tissue leading to many serious side effects such as neurotoxicity. Platinum-induced neurotoxicity can be the result of the following mechanisms: nuclear DNA damage in dorsal root ganglion (DRG) neurons, mitochondrial DNA damage, channelopathy, oxidative stress and mitochondrial dysfunction, and intracellular signaling pathway dysregulation [40,41]. Currently, there are three members of this drug family in use: cisplatin, carboplatin, and oxaliplatin [9]. Carboplatin neurotoxicity seems to be insignificant compared with that of cisplatin and oxaliplatin. It requires a 10-fold higher drug concentration than cisplatin to induce the same cytotoxic effect and predominantly affects the hematopoietic system, while cisplatin and oxaliplatin are primarily associated with CIPN [7,40].

#### 3.2.1. Damage in DRG Neurons

The bodies of sensory neurons located in DRG are believed to be the main target of Pt-based drugs because they need to sustain high metabolism for the maintenance of long axons [41]. Cisplatin and oxaliplatin are apparently substrates of transporters on the neuronal plasma membrane, such as the copper transporters (CTR-1), the organic cation transporters (OCT-1, OCT-2), and the cation and carnate transporters (OCTN-1, OCTN-2) which are likely involved in the influx of Pt-based drugs into DRG neurons [39]. Once inside the cell, platinum compounds reach the nucleus and form DNA adducts, which results in lesions that block DNA replication and transcription [42] that finally leads to accelerated accumulation of unrepaired platinum–DNA adducts and results in cell death [43].

#### 3.2.2. Oxidative Stress and Mitochondrial Dysfunction

Preclinical studies demonstrated that cisplatin forms adducts with mitochondrial DNA at a similar rate as nuclear DNA, which results in inhibition of mtDNA replication, disruption of mtDNA transcription, and morphological changes within mitochondria [44]. However, the main mechanism of platinum-induced toxicity is associated with the overproduction of ROS [45].

#### 3.2.3. Neuroinflammation

Oxaliplatin treatment may trigger an acute inflammatory response that leads to an increase in pro-inflammatory cytokines. Research on rats showed an increase in IL-1β and TNF-α and a decrease in IL-10 and IL-4 in the spinal cord after 25 days of Oxaliplatin treatment [46]. Platinum compounds may also indicate the increase in pro-nociceptive acting chemokines, such as CCL2/CCR2, which are proven to have a significant role in chronic pain in rodents [41]. In addition to changes in cytokine concentrations, the main mechanisms of platinum-related neurotoxicity are also suspected to be supported by the decrease in levels of vitamin E and prealbumin [45].

#### 3.2.4. Enhanced Responsiveness of TRP Channels

Studies in rodents support the idea of TRPV1’s responsibility for the heat-sensitive hyperalgesia and mechanical allodynia in sensory neurons induced by cisplatin, oxaliplatin, bortezomib, and paclitaxel [41].

### 3.3. Neurotoxicity Caused by Vinca Alkaloids

Vincristine alters neuron structure primarily by disrupting the normal assembly and disassembly functions of microtubules, which leads to mitosis block and cell death [47].

### 3.4. Taxane-Induced Neurotoxicity

Taxanes are part of the group of chemotherapeutic agents known as microtubule-stabilizing agents (MTSAs) [48]. Unlike vinca alkaloids, which induce the disassembly of microtubules, paclitaxel inhibits this action, leading to the polymerization of tubulin and the formation of extraordinarily stable and dysfunctional microtubules [49]. Properly functioning microtubules are required for axonal transport and therefore also neuron survival [50].

### 3.5. Bortezomib-Induced Neurotoxicity

Bortezomib-related neurotoxicity may occur through mechanisms of indirect overstabilization of microtubules [51]. Other mechanisms, such as mitochondrial toxicity, endoplasmic reticulum stress in Schwann cells, and inhibition of transcription, transport, and cytoplasmic translation of mRNAs due to accumulation of ubiquitin-conjugated proteins, are also taken into account [52,53].

### 3.6. Thalidomide-Induced Neurotoxicity

For thalidomide, the antitumor mechanisms are suspected to be the cause of its neurotoxicity. TNF-α inhibition and NF-B activation blockade cause neurotrophin and receptor dysregulation, resulting in cell death. The antiangiogenic effect of thalidomide leads to hypoxia and ischemia of nerve fibers [54].

## 4. Current Preventive and Therapeutic Options for CIPN

Prevention and treatment for CIPN are exceedingly challenging because of the varying pathophysiological background of CIPN for different antineoplastic agents [55]. Additionally used so far preventive medications, such as vitamins B and E, glutathione, alpha lipoic acid, acetylcysteine, amifostine, calcium and magnesium, diethyldithiocarbamate, dithiocarbamate, Org 2766, oxcarbazepine, and erythropoietin, were proven to counteract the cancer therapy [56]. Many prevention techniques such as cooling during chemotherapy infusion have been tried to reduce the prevalence of CIPN but without satisfactory results [57]. ASCO and ESMO guidelines suggest checking patients regularly for the development of CIPN and being particularly aware of patients with a high risk of developing neuropathic pain [55].

### 4.1. Pharmacologic Treatment

The only recommended drug that has been clinically proven to reduce pain and other sensory symptoms accompanying CIPN is duloxetine. Although the other options such as tricyclic antidepressants (venlafaxine) and anticonvulsants (gabapentin, pregabalin, and ethosuximide) were tested, the results were controversial [55,57].

### 4.2. Therapies Based on CIPN Mechanisms

#### 4.2.1. Nerve-Protective Therapy

Erythropoietin (EPO), the cytokine produced in kidneys, is proven to have neuroprotective and neurotrophic effects [42]. Studies in rodents have shown that EPO partly prevents the reduction in nerve conduction velocity (NCV) and intraepidermal nerve fibers (IENF) damage caused by docetaxel and cisplatin [58,59,60,61]. EPO may seem to be ideal for peripheral neuropathy but its use is highly contraindicated due to its association with tumor cell proliferation [62].

#### 4.2.2. Ion-Channel-Targeted Therapies

Lidocaine, the inhibitor of voltage-gated sodium channels, is a local anesthetic of amide type also used systematically as an antiarrhythmic drug [63]. A clinical study showed that a single infusion of lidocaine decreased pain in eight out of nine patients with CIPN. The analgesic effect was moderate with a mean duration of 23 days [64,65]. To avoid irritation caused by i.v. injection, lidocaine may be replaced by mexiletine [64]. Examination in mice demonstrated significant reversion of vincristine-induced neuropathic pain after mexiletine treatment [66].

Calcium and magnesium infusions were also postulated as potential CIPN treatment options but research results regarding their effectiveness are inconclusive [55].

Gabapentin and pregabalin are both anticonvulsant drugs whose main mechanism of action is attributed to the α2δ1 subunit of the voltage-dependent calcium channels responsible for modifying the release of neurotransmitters and reducing neuronal excitability [57,67]. This mechanism is responsible for their antiepileptic, analgesic, and sedative properties. Gabapentin may also act by blocking new synapse formation [67]. Despite the apparently identical action mechanisms of gabapentin and pregabalin, the response to those medications varies due to their different pharmacokinetic properties, so that is why patients who do not respond well to gabapentin may experience improvement after pregabalin [68].

#### 4.2.3. Anti-Inflammatory Therapies

The blockage of the nerve growth factor–tyrosine kinase receptor A pathway and treatment with TNF or CCL2 antibodies, as well as increased levels of anti-inflammatory IL-1ra and IL-10, significantly reduce bortezomib- and paclitaxel-induced neuropathic pain [69]. Some studies suggest that the best approach to CIPN treatment is to start with nonsteroidal anti-inflammatory drugs (NSAIDs), eventually followed by an opioid as a second-line agent in the event that NSAIDs fail. However, this strategy requires further research [70].

#### 4.2.4. Neurotransmitter-Based Therapy

Duloxetine and venlafaxine are antidepressants in a group of medicines called selective serotonin and norepinephrine reuptake inhibitors (SSNRIs) that restore the balance of serotonin and noradrenaline in the brain. They are both used for severe depression, anxiety disorder, and other mood problems. Studies revealed that duloxetine is more effective in reducing pain in CIPN than venlafaxine, but regarding the high cost of duloxetine, it may be recommended that venlafaxine would be the first-line medicine in CIPN. In that case, duloxetine would be used in case of no response to venlafaxine [71,72].

Micov et al. investigated the effects of the antidepressant vortioxetine on pain and depression-like behavior in mice with oxaliplatin-induced neuropathy. Vortioxetine reduced mechanical and cold allodynia and depression-like behavior, similar to duloxetine, possibly through increased 5-HT and NA content in the brainstem, suggesting its potential as a treatment option for chemotherapy-induced neuropathy in both pain and depressive symptoms [73].

#### 4.2.5. Antioxidants

Amifostine is an analog of cysteamine that can protect normal cells during chemotherapy by free radical scavenging, donating hydrogen ions to free radicals, depleting oxygen, and binding to active derivatives of antineoplastic compounds [73].

Some randomized trials looked into the neuroprotective effects of amifostine. Indeed, it turned out that premedication with amifostine protected against paclitaxel- and carboplatin-induced sensory neuropathy [74].

Mangafodipir, the superoxide dismutase (SOD) mimetic, is also proven, in both rats and humans, to have protective efficacy against oxaliplatin-associated CIPN. Mangafodipir inhibits oxidative stress by catalyzing the dismutation of superoxide and disarming redox-active iron [75].

## 5. Ozone in the Pain Treatment

### 5.1. Multimodal Mechanism of Action in Pain Management

Most of the potential usage of ozone therapy in various pain treatments stems down from the multimodal mechanism of action of the ozone [76,77,78,79,80]. The most basic mechanism is based on the oxygenation of the infiltrated tissue and the restoration of the cellular redox balance [76,77,78,79,80]. Moreover, ozone modulates the local antioxidant system and, thereby, reduces the inflammatory response, enabling better management of the ischemia/reperfusion processes and promoting fibroblasts and chondrocytes [76,81]. Various studies confirm anti-inflammatory properties along with antigerminal [82,83], antiedema, nerve-regenerating, and analgesic effects [76,82]. Ozone has been found to modulate pain signaling, simultaneously causing alleviation of mechanical allodynia and thermal hyperalgesia [84]. Studies on neuropathic pain caused by chronic constriction injury (CCI) in rats mark the decreased expression levels of upregulated pain signaling components, linked with central sensitization, such as glutamate receptor 6 (GluR6), nuclear factor kappa-B (NF-κB)/p65, IL-1β, IL-6, and TNF-α and phosphorylated NMDA receptor subunit 1 (NR1), NMDA receptor subunit 2B (NR2B), protein kinase C γ (PKCγ), and extracellular signal-regulated kinase (ERK) in the spinal cord after administration of ozone [84]. Ozone administration increases 5′-adenosine monophosphate (AMP)-activated protein kinase (AMPK) activation, simultaneously suppressing increased phosphorylation of NR1, NR2B, PKCγ, and ERK, aforementioned pain-promoting agents and alleviating pro-inflammatory response by suppressing NF-κB activation in macrophages [85,86]. Despite promising premises, contradictory results were obtained in several studies implying the possibility of pain induction by reactive forms of oxygen [87,88].

In addition, ozone has been proven to obtain miorelaxing properties improving mobility of the treated area [89]. The ozone therapy administered in hernia patients modulates the composition of the nucleus pulposus of the intervertebral disk matrix by the reaction with building compounds resulting in the shrinking size of the disk and reduced compression [90,91,92].

### 5.2. Therapeutic Usage of Ozone in Various Pain Syndromes

Ozone in pain treatment was first reported as early as 1960; despite growing interest in the usage of ozone therapy and application in clinical practice, the accessible literature on the topic remains sparse. Due to the lack of the anaphylactic reaction and low infection risk caused by ozone’s antigerminal properties, ozone remains as one of the safest forms of treatment with most of the adverse effects connected to the administration process. Currently, several main areas of the usage of ozone in pain management can be distinguished.

#### 5.2.1. Spinal Pathologies: Lower Back Pain and Disc Hernia

Among spinal pathologies treated with ozone most prevalent are disk hernia [93,94,95,96,97,98,99,100,101,102,103,104,105] and lower back pain [100,102,103,106,107,108] syndromes, although several studies included patients with disk protrusions [95] and spondylosis [95,109]. Ozone was administered in the form of intradiscal, periganglionic, periradicular, and intraforaminal injections or intradiscal, epidural, periradicular, periganglionic, and paravertebral infiltrations. In most research, steroids were implemented along with ozone, although the dosage and application form varied—in the majority, the best results were obtained with the combined treatment [98,104,106]. Most studies reveal promising results within the research group—with a good response rate of up to 78% [110]. Side effects of the treatment are mostly linked to the administration of the ozone and remain reported on a very low scale [111].

#### 5.2.2. Knee Pathologies

Most studies covering the usage of ozone in knee pathologies include osteoarthritis [112,113,114,115,116,117,118,119,120,121,122,123] with up to 71.4% positive results [106]. The application of the ozone involved intra-articular and periarticular infiltrations [112] and injections [106,114,115,122]. In comparison to the usage of hyaluronic acid, no statistically significant differences in efficacy were found with the prevalence of ozone treatment in the early stages of illness, when the inflammation stage is dominant and deformities are less prevalent [106,112,113,114,115,116]. Other knee pathologies treated with ozone constitute posttraumatic arthritis [112,124,125], refractory knee tendinopathies, and patellofemoral chondromalacia, where in most cases improvement was achieved.

#### 5.2.3. Neuralgia, Fibromyalgia, and Nerve Damage

The biggest insight on ozone efficacy has been reported in the field of fibromyalgia, where Hidalgo et al. in two consecutive studies in 2005 and 2013 treated 21 and 36 patients, respectively; Tirelli et al. conducted a study in 2016–2018 on a group of 65 patients; and Moreno-Fernandez in 2019 published the results of the treatment conducted on 20 patients [19,20,21,116,126]. Ozone was administered by autohemotherapy or rectal infusion. In all studies, the majority of patients reported significant improvement, involving a decrease in pain, fatigue, and mental distress. None of the studies reported significant side effects [19,20,21,116,126].

A secondary group of studies on ozone efficacy can be described in the field of postviral nerve damage, with Hu et al.’s study in 2017 on ozone autohemotherapy combined with pharmacological therapy in postherpetic neuralgia, Li et al.’s study in 2022 on acute zoster neuralgia treatment with high-voltage pulsed radiofrequency combined with oxygen–ozone injection, and Zhang et al.’s in 2023 study on the postherpetic neuralgia treated with combined high-voltage pulsed radiofrequency and ozone therapy versus ozone therapy alone [22,23,24]. In the study of Hu et al. patients treated with ozone autohemotherapy obtained significant improvement in comparison to the pharmacological group, meanwhile Zhang et al.’s study results found no statistical differences between the two groups with significant pain improvement in both groups—with an efficacy rate of 73% for combined treatment and 57% for ozone only at 1 year after the treatment [22,23]. Li et al.’s study obtained promising results with a reduction in pain intensity and an improvement in sleep quality, enabling a significant decrease in the implemented pain medication [24].

Ozone therapy in refractory headache has been examined by Clavo et al. in a group of five patients, with significant improvement of the symptoms pre- and posttreatment with best results at 6 months after the treatment (1.1 ± 2.5 on a visual analog scale in comparison to 8.7 ± 0.8 pretreatment) [127].

Rowen and Robins reported major clinical success in the pharmacologically unmanageable regional pain syndrome in 11 years old treated with 120 doses of ozone administered as direct intravenous gas resulting in symptom-free status [25].

Two animal models in rats examining the influence of ozone on nerve regeneration in nerve damage were conducted: Somay et al. in sciatic nerve crush and Ozbay et al. in facial nerve crush [128,129]. Both studies found ozone beneficial in the process of nerve regeneration with the effect on vascular congestion, vacuolization, and myelin thickness [128,129].

### 5.3. Clinical Trials in Ozone Pain Management

Currently, 11 clinical trials involving the usage of ozone in pain treatment are registered, 2 of which cover cancer and chemotherapy-related pain syndromes (Table 1). Most remain of unknown status, with only two actively recruiting, one terminated, and one completed [130,131,132,133,134,135,136,137,138,139]. The most commonly researched condition is back pain, although orofacial pain, osteoarthritis of the knee, diabetic neuropathy, and bladder pain syndrome are also examined under the potential ozone usage [130,131,132,133,134,135,136,137,138,139]. The only completed trial The Effect of Epiduroscopy and Ozone Therapy in Patients With Failed Back Surgery Syndrome (Epiduroscopy), a study completed in 2012 in Sao Paulo, examined the efficacy and safety of ozone therapy administered into the epidural space through epiduroscopy in 40 participants suffering from failed back surgery syndrome. The presented results revealed a 44.0% improvement in the Oswestry Disability Index with a reduction in lumbar and leg pain by 43.7% and 60.9%, respectively, with better results in patients with predominant nonneuropathic pain [139].

A prospective, open-label clinical study by Rania et al. evaluated the effectiveness and safety of intramuscular paravertebral injections of an oxygen–ozone (O2-O3) mixture for cervicobrachial pain in 540 patients. The results showed a significant reduction in pain over time, with all patients becoming pain-free after 1 year of treatment, and no adverse events were observed. The study concludes that the injection of an O2-O3 mixture is a safe and effective treatment option for patients with cervicobrachial pain [140].

## 6. Ozone Therapy in Oncology

Ozone therapy (OT) remains mainly as an element of a multidisciplinary approach in oncology treatment. Its primary role is to alleviate radiotherapy (RT) and chemotherapy (CT) side effects and enhance the effectiveness of conventional treatment with the purpose of obtaining high quality of patients’ life, although several studies pinpoint the possibility of a cancericidal effect on the tumor cells [141,142].

### 6.1. Ozone in Oncology-Action Mechanism

The role of tumor ischemia and hypoxia in cancer progression and development of metastases is well examined. These factors limit the response to CT and RT as well [140].

The high oxygenation potential of ozone determines its possible role in cancer treatment. Similarly, as in arteriopathic patients, ozonated autohemotherapy can increase oxygenation in hypoxic tissues, leading to normoxia [143]. Ozone increases 2,3-biphosphoglycerate in RBS and modifies the hemoglobin dissociation curve by shifting it towards the right side, which is equal to enhancing oxygen saturation. This results in increased oxygen delivery to ischemic tissues. Peroxidation of the erythrocyte membrane improves its flexibility and diminishes blood viscosity. Furthermore, ozone has a vasodilatory effect expressed as inducing the production of vasodilators such as nitric oxide [144].

A pilot study using the technique of polarographic probes has demonstrated changes in tumor oxygenation occurring during ozone therapy. The study included a total of 18 subjects. In all patients, a significant and inverse nonlinear correlation between the increase in oxygenation and the initial tumor pO2 values was revealed [145]. A similar method applied in other studies has demonstrated the negative impact of tumor hypoxia on the survival rate of patients with various types of cancer (sarcoma, tumors in the uterine cervix, and head and neck tumors) [140,146,147].

### 6.2. Animal and Cell Line Models of Ozone Therapy in Cancer Therapy

The direct impact of OT on tumors’ ability to metastasize was examined by Menedez et al. in 2008. Mice had Ehrlich Ascitic Tumor and Sarcoma 37 implanted by the ocular plexus and were treated with ozone by rectal application, which resulted in significant reduction in the metastases in their lungs [148].

The role of ozone in reduction of CT adverse effect was presented in studies on rats. Ozone adjuvant therapy was proved to have protective and antioxidant effects against methotrexate-induced nephrotoxicity, which has oxidative background [149].

Moreover, ozone-oxidative preconditioning may have cardioprotective effect during treatment with doxorubicin. OT contributed to left ventricle morphology preservation, which was accompanied by a decrease in serum indicator of heart failure, pro-BNP level [150].

Furthermore, ozonated water was investigated for direct antitumoral effects, which brought promising outcomes. A study conducted on tumor-bearing mouse models and normal controls showed that ozone affects selectively tumor tissues while remaining harmless to normal tissues. Additionally, OT induces necrosis rather than apoptosis in a possible mechanism of ROS exertion, which is significant to tumor immunity [151].

In combination with cannabidiol (CBD), ozone has demonstrated an antitumor effect on pancreatic ductal adenocarcinoma (PDAC) cell lines. Activation of cannabinoid receptors was found to induce pancreatic cancer cell apoptosis without any effect on the properly functioning pancreas cells which supports CBD usage. The viability of neoplastic cells was remarkably reduced after the addition of oxygen–ozone combination in comparison to CBD alone. Moreover, both ingredients of the mentioned combination were able to significantly affect the expression profile of genes involved in PDAC [152].

In Mendes et al.’s study, the intraperitoneal route of OT administration was used in mice before Lewis’ lung carcinoma inoculation, leading to diminished tumor volume increase [153].

Veterinary studies present OT effectiveness in different types of tumors. Four canine patients with various carcinomas (lymphosarcoma, chondrosarcoma, adenocarcinoma, and osteosarcoma) had ozone therapy applied in treatment cycles rectally, by autohemotherapy and by local infiltration along with CT. The survival rate increased and the quality of life improved in all cases [154].

### 6.3. Clinical Evidence of Ozone Therapy in Cancer Therapy

Ozone therapy in oncology is mostly documented in alleviating various side effects developed in the course of the oncological treatment. Most studies documented inflammatory adverse effects in the course of chemo- or radiotherapy, such as aphthous ulcers, mucositis, proctitis, and enteritis. In the reported cases, ozone treatment leads to the major reduction or complete disappearance of the pain symptoms and attenuates mentioned pathology, possibly by influencing blood flow and oxygenation in hypoxic tissues [148,155,156,157,158]. Moreover, in Yu et al.’s study, in a group of 62 patients with chemotherapeutic enteritis, ozone autotransfusion was found to reduce the blood hypercoagulability [159]. Menéndez et al.’s study involving 70 patients with prostate cancer revealed that the inclusion of ozone in treatment may alleviate RT adverse effects and cause a decrease in prostate-specific antigen (PSA) figures [153].

The usage of medical ozone was also found effective in avascular osteonecrosis of the jaw (ONJ), most commonly caused by bisphosphonates [159,160,161]. In a clinical trial involving 12 patients presenting ONJ symptoms, all patients were relieved of pain, secretions, and halitosis after administration of ozone therapy with complete resolution of ONJ in 8 patients and improvement of the lesions’ persistence in 4 patients, similarly in Agrillo’s study, among 33 patients 18 were completely healed and 10 showed a significant reduction in symptoms and lesions [162,163]. Similar effects were obtained in the case of ONJ co-occurring with mandibular metastasis and local RT [164]. Among the different application of ozone belongs the management of postsurgical ailments, which was found efficient in patients with uterine myoma and endometrial cancer, who received an intravenous infusion of ozonated saline water and experienced alleviation of the side effects of antineoplastic therapy and decreased inflammatory response due to the regulated level of CD16+ lymphocytes [165]. The higher efficacy of wound healing in ozone treatment accelerates the postoperative recovery process, leading to avoiding potential delays in RT and CT [141].

Ozone is also suspected as a potential anticancer agent with several studies establishing it as novel adjuvant therapy. In Clavo et al.’s study on the advanced head and neck cancer patients treated with radiotherapy, patients receiving ozone as adjuvant treatment achieved 2 months advantage in the survival rate in comparison to the adjuvant CT group (6 months) despite worse clinical status—older age and more advanced lymph node involvement than the CT group. The same study indicates RT efficacy, enhancing the potential of ozone due to its capacity of increasing oxygenation [166]. Gaspary reports a case of a 53-year-old male with CT-resistant metastasized stage IV rectal adenocarcinoma presenting the capacity to contain tumor growth observed in examinations after 30 days of ozone with pulsed electromagnetic fields (PEMFs) administration and complete pain alleviation posttreatment despite several adverse effects of the previous CT [26].

Tirelli et al. have also shown, in their 2018 study, strong ozone potential in fatigue management as a supportive care in oncological patients, where, in a group of 50 patients representing various types of neoplasms, 70% of patients reported a significant reduction in fatigue symptoms during therapy or after its termination. Moreover, no adverse effects have been found [167].

## 7. Ozone Therapy in Cancer Pain

Cancer pain, if not treated, may severely disrupt the quality of life, cause functional decline, and increase psychological stress [168]. The mechanisms underlying cancer pain have not been precisely identified yet, but they are different from those responsible for inflammatory and neuropathic pain. Currently, the most likely hypothesis is that cancers generate and secrete mediators that sensitize and activate primary afferent nociceptors in the cancer microenvironment and additionally induce neurochemical reorganization of the spinal cord which leads to spontaneous activity and increased responsiveness [169]. Until recently, conventional science dismissed hypotheses that ozone therapy could be used during anticancer treatment to not only suppress tumor growth but also control symptoms such as severe pain for years due to several flawed experimental designs and very small study samples. Fortunately, now there is some evidence suggesting that ozone therapy (with or without other standard treatments) has various therapeutic effects, including pain management in cancer [15,26]. If the three-stage approach to pain management established by the WHO fails, interventional treatment planning for pain may be a good solution. Interventional pain management can be applied to, among other things, medical ozone injections. Injecting medical ozone into a painful spot is proven to be the most effective way to obtain an analgesic effect using ozone [170]. Ozone therapy may be used in pain palliation in oncology patients because of its ability to increase oxygenation of the tissues of the body and decrease inflammation with antibacterial, antifungal, and antiviral effects [15]. It has also been reported that medical ozone therapy has an antiedema effect on swollen tissue [170].

### 7.1. Ozone Therapy in CIPN

Ozone may have a potentially valuable effect on CIPN because of its ability to modulate oxidative stress, inflammation, and ischemia or hypoxia [171].

#### 7.1.1. Potential Mechanisms

Nrf2 and NF-κB individually affect many signaling cascades to maintain redox homeostasis. Additionally, they interact with each other to further modulate levels of key redox modulators in health and disease [172]. Activation of nuclear transcriptional factor kappa B (NF-κB) results in an inflammatory response and tissue injury via the production of COX2, PGE2, and cytokines. However, moderate oxidative stress activates another nuclear transcriptional factor, nuclear factor-erythroid 2-related factor 2 (Nrf2). Nrf2 then induces the transcription of antioxidant response elements. Several studies have proven the Nrf2 and NF-kB pathways to participate in neuropathic pain mechanisms, and targeting them may have potentially significant therapeutic effects [172,173,174,175]. Ozone has been proven to have an impact on the restoration of the NF-kB/Nrf2 balance [176,177,178]. Exposure of explanted adipose tissue to low ozone concentrations slowed its degradation and induced a concomitant increase in the protein abundance of Nrf2. A study performed on rats with adenine-induced CKD found that ozone therapy reduced tubulointerstitial injury, probably via mediating Nrf2 and NF-kB [179]. In Siniscalco et al.’s study on rats, this time with streptozotocin-induced pancreatic damage, systemic oxygen/ozone administration increased endogenous Nrf2 in pancreatic tissue [180]. The controlled clinical trial in healthy volunteers revealed an immediate increase in levels of Nrf2 after ozone/oxygen exposure [181]. The study of Delgado-Roche et al., aiming to address the role of ozone therapy on the cellular redox state in multiple sclerosis patients, showed an increase in Nrf2 phosphorylation and activation after rectal insufflation with ozone [182]. Based on those studies, it may be speculated that the antioxidant and anti-inflammatory properties of ozone are connected to the activation of Nrf2, which may be crucial for ozone’s role in CIPN management.

TGF-1 is a key regulator of diverse biological processes in many tissues and cell types. Rodent studies revealed that neuropathic pain may be associated with a decrease in TGF-1 expression. The studies in mice revealed that the lack of TGF-1 results in a widespread increase in degenerating neurons and strongly reduces the survival of primary neurons. The deficiency of TGF-1 results in increased neuronal susceptibility to excitotoxic injury, whereas astroglial overexpression of TGF-1 protects adult mice against neurodegeneration in acute, excitotoxic, and chronic injury paradigms [183]. TGF-1 was also reported to alleviate nerve-injury-induced neuropathic pain in rats [184]. Modulation of TGF-1 may be another mechanism underlying ozone’s analgesic properties in CIPN-related pain since ozone is proven to be able to stimulate the synthesis of growth factors, including TGF-1 [185].

#### 7.1.2. Existing Results and Current Trials

A study on rats with streptozotocin-induced diabetic neuropathy showed that rats treated additionally with ozone performed at higher amplitudes of conduction velocity and compound action potential and had higher total antioxidant status, lower total oxidative status, and a lower OS index. This outcome confirmed that ozone partially inhibited the development of drug-induced neuropathy. It also suggests that the preventive properties of ozone are mediated through redox mechanisms [186].

The experimental trial on rats, designed to investigate whether IVF ozone has an analgesic effect on animal models of neuropathic and inflammatory pain, was performed at the Institute for Biomedical Sciences of Pain. The neuropathic pain in rats was produced via separated nerve injury. In this study, IVF injection of ozone at L4-5 proved to be effective in suppression of mechanical allodynia in rats with neuropathic pain. Moreover, the analgesic effects of IVF ozone lasted much longer (>14 days) than other selective molecular target drugs (<48 h), inhibiting or antagonizing at Nav1.8 (A-803467), CXCR4 (AMD3100), mTOR (rapamycin), and histone deacetylase (MGCD0103). Combined use of systemic gabapentin and IVF ozone produced a synergistic analgesic effect in groups with neuropathic pain [187].

In a study on rodents, Ogut et al., aiming to examine the effects of mild-level ozone therapy on sciatic nerve regeneration, revealed an increase in SOD, CAT, GPx, and antioxidant enzymes in plasma and a decrease in MDA levels in groups treated with ozone. Although the study does not directly address CIPN, the regenerating effect of ozone on nerve fibers is worth noting [188].

Studies on ozone therapy in a group of six cancer patients without evidence of tumor relapse but with refractory chronic pelvic pain secondary to cancer treatment (radiotherapy, chemotherapy, surgery, or a combination of them) showed satisfactory preliminary results (Table 2). Following the failure of conventional therapeutic methods, such as anti-inflammatory, co-adjuvants, or opioid drugs, ozone therapy was implemented, and all cases, with the exception of one, demonstrated clinically significant pain reduction. The standard treatment resulted in a visual analog scale score of 7.8 ± 2.1 before O3T, 4.3 ± 3.4 (*p* = 0.049) after 1 month, 3.3 ± 3.7 (*p* = 0.024) after 2 months, and 2.8 ±3.8 (*p* = 0.020) after 3 months [27,28].

The other study conducted on seven patients (two males and five females between 36 and 73 years old) with chronic and painful grade II or III level of CIPN revealed significant improvement in all patients except for one after adjuvant treatment with rectal ozone therapy. The median pain score according to the VAS was 7 (range: 5–8) before ozone treatment, 4 (range: 2–6) at the end of ozone treatment (*p* = 0.004), 5.5 (range: 1.8–6.3) 3 months later (*p* = 0.008), and 6 (range: 2.6–6.6) 6 months later (*p* = 0.008) [29].

Several clinical trials are currently underway to assess the efficacy and additional costs of ozone therapy (Table 1 and Table 2).

A randomized, triple-blind trial will be conducted on 42 patients with any type of cancer and any type of chemotherapy who have CIPN of grade II or higher for more than 3 months. Patients will receive standard care along with 40 rectal insufflation sessions of O3/O2 over the course of 16 weeks: ozone arm (n = 21): concentration of O3/O2 increasing from 10 to 30 g/mL and control–placebo arm (n = 21): concentration of O3/O2 = 0 g/mL. The following main variables will be analyzed at the end of the treatment: “average pain” secondary to CIPN using the Brief Pain Inventory-Short Form (BPI-SF), health-related quality of life (HRQOL) and utilities using the EQ-5D-5L and SF-36 quality-of-life questionnaires, and direct costs. Secondary trial variables include biochemical parameters of oxidative stress and inflammation, the Hamilton scale for anxiety and depression, hyperspectral images, and patient acceptance of a shared decision-making (SDM) tool. The trial with assessments and follow-up is expected to last 36 months in total [128].

Another trial, with 105 patients, aims to assess the clinical effect of adding ozone to standard treatment on HRQOL. The EQ-5D-5L questionnaire will be used to assess HRQOL. Secondary variables include anxiety and depression as measured by the hospital anxiety/depression (HAD) questionnaire, pain as measured by the visual analog scale (VAS), cancer patients’ grade of toxicity as measured by the CTCAE v5.0 scale, the number of invasive procedures required for symptom management, a self-reported percentage of symptom improvement, and biochemical parameters of oxidative stress and inflammation The study is of observational character and is estimated to last 49 months [129].

## 8. Discussion

Chemotherapy-induced peripheral neuropathy belongs to the common side effects of the various antineoplastic agents used in cancer treatment [5,189]. Due to CIPN’s complex character consisting of hyperalgesia, allodynia, hypersensitivity, paresthesia, dysesthesia, pain, and cramping of the extremities, it severely influences patients’ quality of life and can be the reason for the decrease in dosage or treatment termination [31,189]. Simultaneously due to the complex pathomechanism, CIPN tends to remain persistent even after termination of the treatment, sometimes presenting as late offset after treatment completion [189].

A major limitation in CIPN management is the low effectiveness and response rate of the available therapeutic options with little to no preventive compounds available due to their contradictory mechanism to the chemotherapeutic agents [10,55,56,57].

The potential significant role of oxidative stress and redox imbalance in the pathogenesis of CIPN along with the influence of the inflammatory response and dorsal root ganglion damage indicates the strong potential of ozone treatment in CIPN [37,38,41].

Considering numerous reports on the ozone restoration properties on the cellular redox balance and influence on the inflammatory response by decreasing levels of pain signaling agents, such as IL-1β, IL-6 and TNF-α, GluR6, phosphorylated NR1, phosphorylated NR2B, and PKCγ, ozone targets multiple CIPN-induction pathways [85,86]. Moreover, ozone has been found to decrease macrophage inflammatory response by influencing NF-κB/p65, induce a rapid increase in nuclear factor-erythroid 2-related factor 2 (Nrf2) responsible for redox homeostasis maintenance, modulate deficiency of TGF-1 linked with neurodegeneration, and diminish hypersensitization caused by extracellular-signal-regulated kinase (ERK) [85,86,174,175,176,181].

Several reports describe the successful usage of ozone therapy in neuropathic pain syndromes, such as fibromyalgia, postviral neuralgia, and regional pain syndrome as well as various cancer-related pain ailments, with high response rates enabling a decrease in pain medication dosage and high safety profile [15,19,20,21,22,23,116,126,164].

The effectiveness of anticancer ozone treatment in animal models and positive results of adjuvant ozone treatment in people strongly suggest ozone therapy as highly beneficial besides pain relief factor [148,151,153,154,155,156,157,158].

The partial similarity between pathomechanisms of CIPN and other neuropathic pain syndromes indicates high possibility of the successful usage of ozone in CIPN, which is already prevalent in several studies conducted on the action of ozone in CIPN [128,129].

## 9. Conclusions

The scientific evidence proves that ozone, by virtue of its antioxidant, immunomodulatory, and oxygenation properties, can be a valuable complementary therapeutic measure in CIPN treatment. However, its direct effects are not yet well examined and further investigation is warranted in this field.

## Figures and Tables

**Table 1 ijms-24-05279-t001:** Clinical trials on pain treatment with ozone.

Trail Number	Years Conducted	Location	Study Title	Condition	Number Recruited	Ozone Implementation Form	Dosage	Reviewed Data	Status
NCT01172457	2009–2012	University of São Paulo Medical School São Paulo, Brazil	The Effect of Epiduroscopy and Ozone Therapy in Patients With Failed Back Surgery Syndrome	Low Back Pain Failed Back Surgery Syndrome	40	Epiduroscopy	30 mL of ozone at a concentration of 30 μg/mL	Visual analog pain scale—VAS	Completed
NCT00832312	2009–2016	Ben Gurion University Beer Sheva, Israel	Intraarticular Ozone Therapy for Pain Control in Osteoarthritis of the Knee	Osteoarthritis of the Knee	20	Intraarticular injections	10 cc of an ozone–oxygen mixture with an ozone concentration of 10,000 μg/L (10 μg/mL)	Pain control	Terminated
NCT01709058	2012–2015	INRCA Hospital, via della Montagnola, 81 Ancona, Ital	Study on the Effects of Oxygen-ozone Therapy on Back Pain in Subjects Aged 65 or Older	Back Pain	130	Intramuscular/paravertebral injections	5–20 mL of oxygen–ozone to each point, at a concentration of 10–20 μg/mL, for a total volume of 40 mL	Oswestry Disability Index (ODI)	Unknown
NCT02997410	2014–2017	Tamer Celakil, Istanbul University	Ozone Therapy for Masticatory Muscle Pain (OTMMP)	Orofacial Pain Temporomandibular Joint Disorders	60	Dental treatment with occlusal splint	Not disclosed	Pressure pain threshold measurement (PPT) using a Pressure Algometry; pain scores on the Visual Analog Scale	Unknown
NCT03056911	2017–2019	Sakarya University Research and Training Hospital Sakarya, Turkey	Clinical Effects of Ozone Therapy in Cervical Disc Hernia	Neck Pain	43	Chemonucleolysis	Not disclosed	Visual analog scale (VAS) score for pain	Unknown
NCT04789135	2020–2021	Distal Nefrologia e Urologia Jacareí, SP, Brazil	Evaluation of Response to Use of Intravesical Ozone Gas in Interstitial Cystitis/Bladder Pain Syndrome	Interstitial Cystitis, Chronic Bladder Pain Syndrome	50	Direct instillation into the bladder of ozone gas via urethral catheter	Concentration of 20–60 μg/mL	O’Leary Sant Symptom and Problem Index questionnaire	Active, not recruiting
NCT04562493	2020–2021	Wael Fathy Hassan Banī Suwayf, Egypt	Comparative Effect of Transforaminal Injection of Magnesium Sulphate Versus Ozone Therapy on Oxidative Stress Biomarkers in Lumbar Disc Related Radicular Pain	Effect of Drug	90	Transforaminal epidural injection	Not disclosed	Oxidative stress biomarkers: Glutathione (GSH) and superoxide dismutase (SOD); Pain relief: Numeric Rating Scale Disability improvement: Numeric Rating Scale	Unknown
NCT04299893	2020–2023	Complejo Hospitalario Materno Insular Las Palmas De Gran Canaria, Las Palmas, Spain Hospital Universitario de Gran Canaria Dr. Negrín Las Palmas De Gran Canaria, Las Palmas, Spain	Ozone Therapy in Chemotherapy-induced Peripheral Neuropathy: RCT (O3NPIQ)	Chemotherapy-induced Peripheral Neuropathy Pain, Neuropathic Pain Syndrome	42	Rectal insufflation	Concentration of O3/O2 increasing from 10 to 30 μg/mL	Brief Pain Inventory-Short Form (BPI-SF)	Recruiting
NCT05000463	2022	Emad Zarief Kamel Said Assiut, Egypt	Ozone Therapy in Patients With Diabetic Neuropathy	Chronic Pain	60	Nerve injection	Ozone/oxygen mixture (25 μg/mL)	Visual analog scale of pain (VAS)	Not yet recruiting
NCT05417737	2022–2026	Dr. Negrín University Hospital Las Palmas, Spain	Patients Referred to the Chronic Pain Unit for Palliative Treatment With Ozone Therapy Between 2022 and 2025	Radiation Toxicity Chemotherapeutic Toxicity Chemotherapy-induced Peripheral Neuropathy Delayed Wound Healing Chronic Pain Refractory Pain	105	Systemic and/or local ozone administration	Not disclosed	5-level, 5-dimension EuroQol (EQ-5D-5L) questionnaire	Recruiting
NCT05291715	2022–2023	Cairo University-Faculty of Dentistry Cairo, Manial, Giza, Egypt	The Effect of Ozone Therapy on Pain Perception After Free Gingival Graft Surgery in Patients With Mucogingival Defects	Open Wound of Palate Without Complication Pain, Postoperative Free Gingival Grafts Mucogingival Defects Gingival Recession Donor Site Patient Satisfaction	24	Ozone generator device	Not disclosed	Visual analog scale (VIS)	Recruiting

**Table 2 ijms-24-05279-t002:** Comprehension of reported studies on neuropathic and cancer-related pain treated with ozone.

Author	Year	Condition	Way of Administration	Dosage	Therapeutic Effect
Javier Hidalgo-Tallo’n et al. [19]	2012	Fibromyalgia	Rectal insufflation	8 mg (200 mL of gas, at a concentration of 40 μg/mL), 5 days a week in the first week, 2 times a week from weeks 3–6, weekly from weeks 7–12	Significant decrease in FIQ total scores, significant improvement in depression scores and in the Physical Summary Score of the SF-12
U. Tirelli et al. [20]	2019	Fibromyalgia	Autohemotransfusion, rectal insufflation	According to SIOOT protocols, twice a week for 1 month, then twice a month as maintenance therapy	Significant improvement in 45 of 65 patients (70%), no side effects reported
Moreno-Fernández et al. [21]	2019	Fibromyalgia	Autohemotherapy	150 mL of O_3_ in 150 mL of blood at a concentration of 30–60 μg/mL, 10 sessions (2 sessions/week) for 7–10 min	Significant decrease in tender points and FIQ, improvement in sleep and mental alertness, a moderate increase in serotonin levels, an important decrease in LP and PC, decrease in ROS
Jian-Feng Zhang et al. [22]	2022	Postherpetic neuralgia	Ozonated water injection through the inner cannula	10 mL (truncal OT)/7 mL (facial OT) of mixture at a concentration of 11.5 µg/mL with the infusion speed of 3 mL/min for truncal OT and 2 mL/min for facial OT, once a day, Monday to Friday for 1–2 weeks	Significant improvement in pain and tactile sensation
Bin Hu et al. [23]	2018	Postherpetic neuralgia	Autohemotherapy	40 mL of ozone in 200 mL of blood at a concentration of 30 μg/mL, transfused back within 15 min, 3 times/week for 2 weeks	Significant improvement in the VAS, MPQ, PGIC, and WHOQOL-BREF
Li-Mei Li et al. [24]	2022	Acute zoster neuralgia	O_2_/O_3_ injection	-	Improved pain intensity and sleep quality
Robert Jay Rowen, Howard Robins [25]	2019	Complex regional pain syndrome	Direct intravenous gas (DIV)	At the beginning: 5 cc of gas at a concentration of 55 μg/cc, gradually increased to 30 cc at 55 μg/cc 5 times/week for 26 weeks (120 sessions in total)	Complete remission of pseudoseizures and pain
J. F. Pollo Gaspary et al. [26]	2020	Stage IV rectal adenocarcinoma with liver and lung metastases	Rectal insufflation Ozonated Olive Oil inhalation (OT combined with PEMF)	Rectal route: 8 mg/day, 5 times/week Ozonated Olive Oil: 40 min/day, 5 times/week	Improvement in well-being, autonomy, and pain control, pause in tumor growth despite more than 60 days without using classic treatment
Bernardino Clavo et al. [27,28]	2020	Chronic pelvic pain secondary to cancer treatment	Rectal insufflation, intravesical insufflation/instillations of ozonated water, vaginal insufflation/vaginal washing with ozonated water	Rectal insufflation: 180 mL/session at the beginning, increase up to max. of 300 mL/session if tolerated, initial concentration: 10 µg/mL, increased 5 µg/mL every two sessions up to max. of 30 µg/mL	Significant decrease in pain in five of six patients, improvement in associated symptoms (vaginal dryness, hematuria, rectal or vaginal wounds, tenesmus, and the number of bowel movements per day)
Bernardino Clavo et al. [29]	2022	Pain secondary to grade II or III CIPN	Rectal insufflation	Initial concentration: 10 μg/mL, increased by 5 μg/mL every two sessions until max. of 30 μg/mL, the gas volume started at 180 mL/session and was slowly increased in successive sessions (depending on patient tolerance of bowel bloating) up to max. of 300 mL/session if tolerated	Clinically relevant pain improvement

Note: FIQ—Fibromyalgia Impact Questionnaire; SIOOT—Scientific Society of Oxygen Ozone Therapy; VAS—Visual Analog Scale; MPQ—McGill Pain Questionnaire; PGIC—Patients’ Global Impression of Change; WHOQOL—BREF—World Health Organization Quality of Life.

## Data Availability

Not applicable.

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
