# Peer review of "Ozone in Chemotherapy-Induced Peripheral Neuropathy—Current State of Art, Possibilities, and Perspectives"

_ijms, 2023, doi:10.3390/ijms24065279_

Round 1

Reviewer 1 Report

I was pleased to have the opportunity to revise the manuscript entitled “Ozone in Chemotherapy-Induced Peripheral Neuropathy—Current State of Art, Possibilities and Perspectives” which aims at reviewing the literature on the role of Ozone therapy in oncological patients affected by chemotherapy-induced neuropathy. Overall, the present review is well structured and the topic is comprehensively covered. The most meaningful findings from the literature are well described, exhaustive and consistent. 

Please find below few minor comments.

Introduction section

-The authors state that” Unfortunately medications commonly used for the management of other neuropathic pain syndromes do not bring satisfactory results in CIPN..” Examples of these medications should be given. 

-“ Therapies that take into account OS role in CIPN pathogenesis appear to be promising...”. This line is unclear and need to be reformulated.

-Clinical manifestations of CINP are only briefly described. I suggest to expand this section. 

Neurotransmitter-based therapy section

-The authors should consider the possibility of discussing the potential role of Vortioxetine in the CINP. Indeed, while there is growing evidences of its effectiveness in the treatment of mood disorders and chronic pain conditions, its use in CINP has been demonstrated to be effective in preventing and treating CINP in an animal study from Micov MA et al. (Vortioxetine reduces pain hypersensitivity and associated depression-like behavior in mice with oxaliplatin-induced neuropathy).

Spinal pathologies: Lower Back Pain and Disc Hernia 

Ozone therapy has also been effectively used in the management of cervicobrachial pain as recently demonstrated by Rania V. et al in their study (Oxygen–Ozone Therapy in Cervicobrachial Pain: A Real-Life Experience). The authors may discuss the relevant findings in this section.

-A referral to Tables 1 and 2 should be added appropriately in the text.

-Few English and typo mistakes.

Author Response

I was pleased to have the opportunity to revise the manuscript entitled “Ozone in
Chemotherapy-Induced Peripheral Neuropathy—Current State of Art, Possibilities and
Perspectives” which aims at reviewing the literature on the role of Ozone therapy in
oncological patients affected by chemotherapy-induced neuropathy. Overall, the present
review is well structured and the topic is comprehensively covered. The most meaningful
findings from the literature are well described, exhaustive and consistent.
Please find below few minor comments.
Introduction section
-The authors state that” Unfortunately medications commonly used for the management of
other neuropathic pain syndromes do not bring satisfactory results in CIPN..” Examples of
these medications should be given.
Change 1) Examples of drugs have been added in the introductory part. This is
especially true of opioids and some antidepressants.
-“ Therapies that take into account OS role in CIPN pathogenesis appear to be
promising...”. This line is unclear and need to be reformulated.
Change 2) This passage has been thoroughly reformulated. The point is that
oxidative stress as one of the possible explanations for the effectiveness of some
antidepressants may set the direction of research into new drugs.
-Clinical manifestations of CINP are only briefly described. I suggest to expand this
section.
Change 3) This section has been expanded. It was indicated which symptoms
appear first, and which of them are the most characteristic.
Neurotransmitter-based therapy section
-The authors should consider the possibility of discussing the potential role of Vortioxetine
in the CINP. Indeed, while there is growing evidences of its effectiveness in the treatment
of mood disorders and chronic pain conditions, its use in CINP has been demonstrated to
be effective in preventing and treating CINP in an animal study from Micov MA et al.
(Vortioxetine reduces pain hypersensitivity and associated depression-like behavior in
mice with oxaliplatin-induced neuropathy).
Change 4) Based on the indicated paper, information about vortioxetine has been
added.
Spinal pathologies: Lower Back Pain and Disc Hernia
- Ozone therapy has also been effectively used in the management of cervicobrachial pain
as recently demonstrated by Rania V. et al in their study (Oxygen–Ozone Therapy in

Cervicobrachial Pain: A Real-Life Experience). The authors may discuss the relevant
findings in this section.
Change 5) Based on the indicated paper, information about ozone therapy
demonstrated effectiveness in the management of cervicobrachial pain has been
added.
-A referral to Tables 1 and 2 should be added appropriately in the text.
Change 6) Table references have been incorporated into the text.
-Few English and typo mistakes.
Minor changes) The text was revised, minor punctuation and language errors were
corrected without changing the meaning and content of the expressions used.

Reviewer 2 Report

The paper sounds interesting and has the potential to publish in the above journal after the authors address the following major comments:

-  The abstract section is not well written. It looks like a review of what has been done in the field and it should belong to the part of the introduction. In this section, the authors should present what their innovation is in the current study, what the results are, and what methods they used to achieve these results.

- Until page 13 I can't see the research question that the authors should discuss and solve. I can't see the method that the authors used in this paper.

After page 13 the authors jump directly to the Discussion section. 

Please organized the paper as a research paper. 

After the abstract and the introduction section, please present the main problem that you are going to deal with and solve, what we call "the question of the research". After that, you should present the tools and methods that you apply in order to solve the above research. Then present the results, then discuss and analysis of the results and finally present the conclusion section.

Good luck. 

Author Response

- The abstract section is not well written. It looks like a review of what has been done in the
field and it should belong to the part of the introduction. In this section, the authors should
present what their innovation is in the current study, what the results are, and what
methods they used to achieve these results.
Change 7) The abstract has been rebuilt. Missing methodological information has
been added.

- Until page 13 I can't see the research question that the authors should discuss and solve.
I can't see the method that the authors used in this paper.

After page 13 the authors jump directly to the Discussion section.

Please organized the paper as a research paper.

After the abstract and the introduction section, please present the main problem that you
are going to deal with and solve, what we call "the question of the research". After that,
you should present the tools and methods that you apply in order to solve the above
research. Then present the results, then discuss and analysis of the results and finally
present the conclusion section.

ood luck.
The review is narrative in its nature as the available data on ozone relate to pain
management rather than CIPN specifically. Nevertheless, these data may be a
premise for further research. However, the indicated methodological shortcomings
became the basis for rebuilding the article to clearly indicate what its purpose is.
We regret that the structure of the article did not include this information properly
presented.
Change 8) A methodological section has been added. The research question and
the source of data were indicated.

Reviewer 3 Report

I have reviewed the article and a great job has been done but it could be improved in the following aspects:

·       *  References in the text are not all included in the same way. Some appear before or after the full stop.  Example: Line 23 and line 38

·     *    The acronym ROS is in the introduction and not in the abstract, which is where it first appears. Example: Line 12 and line 33

·      *   It is better to put it this way: Line 364: [127-129]

·    *     Better to put "Menendez et al" than in another study. Line 446

·       *  This text "health-related quality of life (HRQOL)"  is repeated in line 599 and 606.

·     *    Tables are not mentioned in the text.

-Numerous research articles and the clinical trials that have been carried out on ozone have been included in this review, but in the discussion and conclusions sections they do not highlight the possibilities and benefits of ozone which is what they indicate in the title.  These sections should be redrafted to be more in line with what is stated in the title. 

-Table 1 has too many columns and too many long rows. Columns that give little information, e.g. year, location..., should be omitted.  And in the case of the rows, it is noted that acronyms and their meaning are listed.  It is more normal to put the acronym in the row and the acronym and its meaning at the end of the table. Example: VAS: Visual analogue scale Pain

I hope that the changes I indicate will help you in your review article.

Author Response

Hello, 

thank you for finding time to give attention to our article and pinpoint the  suggestions and pointers in the review process. We found all of the provided comments very helpful and tried our best to apply all of the given remarks accordingly.

References in the text are not all included in the same way. Some appear before or after the full stop. Example: Line 23 and line 38

 Change 9) Throughout the article, references have been checked and presented after a period sign (dot).

  • * The acronym ROS is in the introduction and not in the abstract, which is where it first appears. Example: Line 12 and line 33

  Change 10) Throughout the article, the acronyms have been revised so that they are now always explained the first time they are used.

  • * It is better to put it this way: Line 364: [127-129]

Change 11) Throughout the article, reference scopes have been checked so that they are used this way everywhere.

  • * Better to put "Menendez et al" than in another study. Line 446

Change 12) Throughout the article, expressions referring to individual studies were checked so that now the names of the authors are mentioned.

  • * This text "health-related quality of life (HRQOL)" is repeated in line 599 and 606.

 Change 13) Unnecessary repetitions in this part of the text have been eliminated.

  • * Tables are not mentioned in the text.

 Change 14) Table references have been added.

Round 2

Reviewer 2 Report

 I have read the paper again and confirm that the authors revised the paper according to my comments. 

One issue should be solved before the next step: Please send the paper to English editing.